# Efficient Bayesian Additive Regression Models For Microbiome Studies

**Tinghua Chen**
Pennsylvania State University
tuc579@psu.edu

**Michelle Pistner Nixon**
Pennsylvania State University
Geisinger
mpnixon@geisinger.edu

**Justin D. Silverman**
Pennsylvania State University
justinsilverman@psu.edu

## Abstract

Bayesian multinomial logistic-normal (MLN) models have gained popularity due to their ability to account for the count compositional nature of microbiome data. Recently, we developed a computationally efficient and accurate approach to inferring MLN models with a Marginally Latent Matrix-T Process (MLTP) form: MLN-MLTPs. However, previous research on MLTPs has been restricted to linear models or a single non-linear process. This article addresses this deficiency by introducing a new class of MLN Additive Gaussian Process models (*MultiAddGPs*) for deconvolution of overlapping linear and non-linear processes. We show that MultiAddGPs are examples of MLN-MLTPs and derive an efficient Collapse-Uncollapse (CU) sampler for this model class. Through simulation studies, we show that MultiAddGPs accurately and efficiently decompose overlapping effects in microbiota data, which provides a powerful tool for analyzing complex count compositional datasets.

## 1 Introduction

Dysregulation of human-, animal-, and even plant-associated microbial communities (microbiota) are known to cause disease [3, 7, 17, 29, 8]. In humans, alterations of microbiota play a causal role in obesity [33, 20], inflammatory bowel disease [18, 9, 19], and even cancer [27, 16]. As a result, many researchers study how dietary, host physiologic, and environmental factors influence the relative abundance of different bacterial taxa in microbiota. These factors can have linear or non-linear effects on community structure [6, 26, 28]. Overall, flexible statistical methods are needed to disentangle linear and non-linear effects on microbiota.

Beyond the biological complexity of microbiota, limitations of the measurement process further complicate analyses. These data are typically represented as a $D \times N$ count table $Y$ with elements $Y_{dn}$ denoting the number of DNA molecules from taxon $d$ observed (sequenced) in sample $n$. The size of one sample (the sequencing depth; $\sum_{d=1}^{D} Y_{dn}$) is typically arbitrary and unrelated to the total microbial load in the system [34]. As a result, many authors call these data compositional, reflecting the idea that the data only provide information about the relative abundances of the different taxa within each community [10, 24, 23]. Bayesian Multinomial Logistic Normal (MLN) models have gained popularity due to their ability to address challenges in the measurement process. [30, 1, 12, 31, 32]. The multinomial is used to model uncertainty due

Workshop on Bayesian Decision-making and Uncertainty, 38th Conference on Neural Information Processing Systems (NeurIPS 2024). This workshop paper represents a preliminary version of our ongoing work, with an extended version currently under review for journal publication [5].

to random counting, while the logistic normal captures the extra-multinomial variability typically seen in these data [32]. Unlike the more well-known Dirichlet distribution, the logistic-normal has a rich covariance structure which allows modeling both positive and negative covariation between taxa [2, 31]. The logistic-normal is also self-conjugate (as it is multivariate normal under a suitable log-ratio transformation), allowing for a wide variety of models to be built in the latent simplex space. However, the multinomial and the logistic-normal are not conjugate, making inference of these models computationally challenging or even intractable.

Recent advances have made Bayesian MLN models practical for microbiota analyses [4, 11, 21, 1, 31, 12]. Yet scalability limits those methods. For example, the sampler used in [31] took more than four hours on a high-performance cluster to analyze a dataset of approximately one thousand samples, yet only ten taxa. More recently, we proved that a wide variety of Bayesian MLN models, including generalized linear models and generalized Gaussian process regression models, share a common marginal form called a Latent Matrix-T Process (LTP) [32]. We showed that a Laplace approximation to this marginal form was extremely accurate, leading to an efficient and accurate approximate inference procedure called the *Collapse-Uncollapse (CU)* sampler. Our result demonstrated that this approach is often 4-5 orders of magnitude faster than HMC-based methods with minimal error in posterior calculations [32].

Despite these advances, there remains a dearth of tools for disentangling the effects of multiple measured factors on microbiota. Recently, [6] proposed an additive Gaussian process framework to address this need. Yet their approach assumed the data was transformed Gaussian, ignoring count compositional nature of these data. Moreover, our prior work with MLTPs was limited to factors that have a linear effect on microbial composition (generalized linear models) or a single factor that had a nonlinear effect (generalized Gaussian process regression models).

This article addresses the limitations of prior methods and develops a flexible, and computationally efficient approach to disentangling both linear and nonlinear effects on microbiota. As in [6], our approach is based on a class of additive Gaussian process regression models. Unlike [6], we do not assume that the data is transformed Gaussian and instead prove that Bayesian MLN Additive Gaussian Process Models are also MLTPs. Using those results, we extend the CU sampler to this class of models.

We organize this article as follows. Section 2 presents a multinomial logistic-normal generalized additive Gaussian process regression (MultiAddGP) model and proves that this model is part of the Marginally LTP class. Sections 3 demonstrate our approach through application to simulated microbiome data. Finally, we conclude with a discussion in Section 4.

## 2  Methods

To facilitate additive linear and nonlinear modeling within a Bayesian MLN framework, this article introduces Multinomial Logistic Normal Additive Gaussian Process Models (MultiAddGPs). In this section, we first present the model and demonstrate that MultiAddGPs are a specific type of MLTP model (a comprehensive review of MLTP models and Collapse-Uncollapsed (CU) sampler are provided in Appendix A). Following this, we outline the process for conducting posterior inference using an extended CU sampler in MultiAddGPs. Finally, we describe the identification problem in our model.

### 2.1  Multinomial Logistic Normal Additive Gaussian Process Models (MultiAddGP)

Let $\mathbf{Y}_{\cdot n}$ denote a $D$-vector of observed data, $\mathbf{X}_{\cdot n}$ denote a $Q_0$-vector of covariates to model linearly, and each $\mathbf{Z}_{\cdot n}^{(k \in \{1,\ldots,K\})}$ denote a $Q_k$-vector of covariates to be modeled with distinct non-linear functions. The MultiAddGPs models have the following form:

$$\mathbf{Y}_{\cdot n} \sim \text{Multinomial}(\mathbf{\Pi}_{\cdot n}) \tag{1}$$

$$\mathbf{\Pi}_{\cdot n} = \phi^{-1}(\mathbf{H}_{\cdot n}) \tag{2}$$

$$\mathbf{H}_{\cdot n} \sim N(\mathbf{F}_{\cdot n}, \mathbf{\Sigma}) \tag{3}$$

$$\mathbf{F} = \mathbf{B}\mathbf{X} + \sum_{k=1}^{K} \mathbf{f}^{(k)}(\mathbf{Z}^{(k)}) \tag{4}$$

with priors $\mathbf{B} \sim N(\mathbf{\Theta}^{(0)}, \mathbf{\Sigma}, \mathbf{\Gamma}^{(0)})$, $\mathbf{f}^{(k)} \sim \mathsf{GP}(\mathbf{\Theta}^{(k)}, \mathbf{\Sigma}, \mathbf{\Gamma}^{(k)})$, and $\mathbf{\Sigma} \sim \mathsf{InvWishart}(\mathbf{\Xi}, \zeta)$. As in the Appendix A, $\phi$ denotes any log-ratio transform from $\mathbb{S}^D$ to $\mathbb{R}^{D-1}$. $\mathbf{\Sigma}$ is a $D-1 \times D-1$ covariance matrix. For the matrix-normal prior on the linear term, $\mathbf{\Theta}^{(0)}$ is the mean matrix and $\mathbf{\Gamma}^{(0)}$ is a $Q_0 \times Q_0$ covariance matrix representing covariance in the parameters of the $Q_0$ covariates. The terms $\mathbf{\Theta}^{(k)}$ and $\mathbf{\Gamma}^{(k)}$ in the $K$ matrix-normal process priors echo their linear counterparts but are functions (e.g., mean and kernel functions) rather than fixed dimensional matrices. As we will show through simulated data analyses in Section 2.2, this is a very flexible form of model that can be used in a wide range of additive linear and non-linear modeling tasks.

## 2.2 Posterior Estimation in MultiAddGPs

We use MLTP theory to sample from the posterior of MultiAddGPs: $p(\mathbf{H}, \mathbf{B}, \mathbf{f}^{(1)}, \ldots, \mathbf{f}^{(K)}, \mathbf{\Sigma} \mid \mathbf{Y})$. Appendix A provides a review of MLTP theory. In brief, we sample the posterior in two steps. First, we use a Laplace approximation to sample $p(\mathbf{H} \mid \mathbf{Y})$. This is called the collapsed step of the CU sampler [32]. In Appendix B we prove $p(\mathbf{H} \mid \mathbf{Y})$ is a LTP and derive its parameters. We can then use results from [32] which provide efficient algorithms for obtaining MAP estimation and forming a Laplace approximation. For each sample of $\mathbf{H}$ from the approximate posterior, we obtain a corresponding sample from the conditional posterior $p(\mathbf{B}, \mathbf{f}^{(1)}, \ldots, \mathbf{f}^{(K)}, \mathbf{\Sigma} \mid \mathbf{H})$. This is called the uncollapse step of the CU sampler [32]. Before describing subtleties of our uncollapse algorithm, we must first clarify the definition of $\mathbf{F}, \mathbf{f}_1, \ldots, \mathbf{f}_K$.

Up to this point, we have not distinguished between the set of points $n \in \{1, \ldots, N\}$ at which we have observed data $\mathbf{Y}$ and the potentially different set $n^* \in \{1, \ldots, N^*\}$ at which we want to evaluate the functions $\mathbf{F}, \mathbf{f}_1, \ldots, \mathbf{f}_K$. In what follows, we use the symbols $\mathbf{F}, \mathbf{f}_1, \ldots,$ and $\mathbf{f}_K$ to denote the evaluation of corresponding infinite-dimensional functions at the set of evaluation points $\{1, \ldots, N^*\}$, i.e., they are each $D \times N^*$-dimensional random matrices. In contrast, all other random matrices (e.g., $\mathbf{H}$) represent their corresponding infinite-dimensional analogues evaluated at the set of observed points ($\{1, \ldots, N\}$).

Sampling from the uncollapsed form starts by obtaining samples from $p(\mathbf{F}, \mathbf{\Sigma} \mid \mathbf{H})$. Conditioning on $\mathbf{H}$ and marginalizing over $\mathbf{f}^{(1)}, \ldots, \mathbf{f}^{(K)}$ in the MultiAddGP model results in a Bayesian matrix-normal process model with likelihood $\mathbf{H}_{\cdot n} \sim N(\mathbf{F}_{\cdot n}, \mathbf{\Sigma})$ and priors:

$$\mathbf{F} \sim \mathsf{GP}\left(\mathbf{\Theta}^{(0)}\mathbf{X} + \sum_{k=1}^{K} \mathbf{\Theta}^{(k)}(\mathbf{Z}^{(k)}), \ \mathbf{\Sigma}, \ \mathbf{X}^T\mathbf{\Gamma}^{(0)}\mathbf{X} + \sum_{k=1}^{K} \mathbf{\Gamma}^{(k)}(\mathbf{Z}^{(k)})\right)$$

$$\mathbf{\Sigma} \sim \mathsf{InvWishart}(\mathbf{\Xi}, \zeta).$$

This is the same model discussed in [32]: samples from $p(\mathbf{F}, \mathbf{\Sigma} \mid \mathbf{H})$ can be obtained via methods described in Appendix C of that article.

Finally, conditioned on samples of $\mathbf{F}$ and $\mathbf{\Sigma}$, we obtain samples of each $\mathbf{f}^{(k)}$. Inspired by the backfitting algorithm used for estimation in generalized additive models [14], we developed a *backsampling* algorithm for this task. The backsampling proceeds by iteratively sampling $p(\mathbf{B} \mid \mathbf{F})$, $p(\mathbf{f}_1 \mid \mathbf{F}, \mathbf{B})$, $p(\mathbf{f}_2 \mid \mathbf{F}, \mathbf{B}, \mathbf{f}_1)$, …, and then $p(\mathbf{f}^{(K)} \mid \mathbf{F}, \mathbf{B}, \mathbf{f}_1, \ldots, \mathbf{f}^{(K-1)})$. For brevity, we leave a description of this algorithm to Appendix B and C.

## 2.3 Model Identification

Identifiability is a well-known challenge in function decomposition models, such as generalized additive models [13]. Common approaches to address this issue include imposing sum-to-zero constraints on the functions (e.g., $\int \mathbf{f}^{(k)} d\mathbf{Z}^{(k)} = \mathbf{0}$) [13], or modifying kernel functions to enforce identifiability [22]. For simplicity, in this work, we adopt the sum-to-zero constraint by centering posterior samples of each $\mathbf{f}^{(k)}$ as $\mathbf{f}^{(k)} - \mathbf{1} \cdot \mathrm{mean}(\mathbf{f}^{(k)})$.

By leveraging the MultiAddGP model and the CU sampler, we provide a computationally efficient framework for disentangling linear and nonlinear effects in microbiota data while addressing the compositional nature of count data. In the section 3, we demonstrate the model's effectiveness through simulated studies, designed to capture the key challenges highlighted earlier.

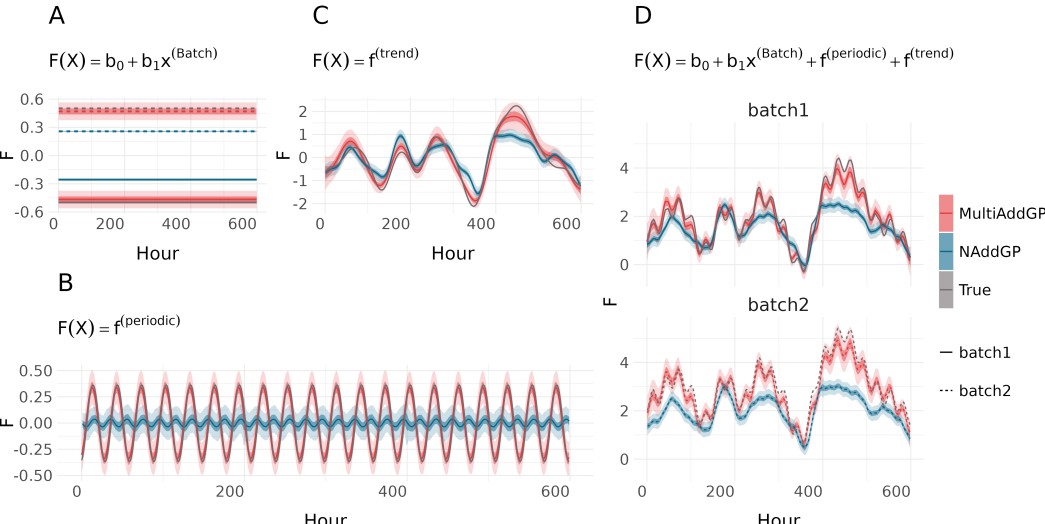

Figure 1: **MultiAddGPs successfully decompose simulated microbiome time-series.** The NAddGP model is identical to the MultiAddGP model but ignores uncertainty due to counting by modeling the observed data as transformed Gaussian. Panels A, B, and C represent individual decomposed components associated with each covariate. Panel D illustrates the cumulative effect of all components. Note: This figure is also included in an extended version of this work currently under review for journal publication [5].

## 3   Empirical Result

We simulated a suite of longitudinal studies of microbiota with varying numbers of taxa $D \in \{3, \ldots, 100\}$ and samples $N \in \{20, \ldots, 1000\}$. We simulated microbial composition influenced by batch effects, daily periodicity (e.g., circadian rhythm; [15]), and longer-term trends. Full simulation details are provided in Appendix D.

In Figure 1 we show a small simulation $D = 4$ and $N = 600$ for ease of visualization. We use $t_n$ to denote the time at which sample $n$ was obtained. For inference, we specify a MultiAddGP model $\mathbf{F}_n = \mathbf{b}_0 + \mathbf{b}_1 x_n^{(\text{batch})} + \mathbf{f}^{(\text{periodic})}(t_n) + \mathbf{f}^{(\text{trend})}(t_n)$ as follows. For covariates, we set $\mathbf{X}_{\cdot n} = [1 \ x_n^{(\text{batch})}]^T$, $\mathbf{Z}_{\cdot n}^{(\text{periodic})} = t_n$, and $\mathbf{Z}_{\cdot n}^{(\text{trend})} = t_n$. For priors we set $\mathbf{B} = [\mathbf{b}_0 = 2.7; \mathbf{b}_1 = 1]$. Both $\mathbf{f}^{(\text{periodic})}$ and $\mathbf{f}^{(\text{trend})}$ were given matrix-normal process priors with mean function $\mathbf{\Theta}^{(k)} = \mathbf{0}$. A periodic kernel $K_{\text{period}}(t, t') = \sigma_{\text{period}} \exp\left(-\frac{2\sin^2\left(\frac{\pi|t-t'|}{p}\right)}{\rho_{\text{period}}^2}\right)$ was used in the prior for $\mathbf{f}^{(\text{periodic})}$ and a squared exponential $K_{\text{trend}}(t, t') = \sigma_{\text{trend}}^2 \exp(-\frac{(t-t')^2}{2\rho_{\text{trend}}^2})$ was used for $\mathbf{f}^{(\text{trend})}$. Hyperparameters $\Omega = \{\sigma_{\text{period}}, \rho_{\text{period}}, p, \sigma_{\text{trend}}, \rho_{\text{trend}}\}$ were selected using MML estimation.

For comparison, we created a nearly identical model that ignored uncertainty due to counting and assumed the data was transformed Gaussian. We implemented this model by setting $\mathbf{H}_{\cdot n} = \phi(Y_{\cdot n} + 0.5)$ and proceeding with the uncollapse step of MultiAddGPs directly; skipping sampling the posterior of the collapsed form. We call this model the *Normal Additive GP (NAddGP)* model.

We compared posterior estimates from the MultiAddGP and NAddGP models to emphasize the importance of modeling uncertainty due to counting. Figure 1 shows that the MultiAddGP almost perfectly recovered the true decomposition whereas the NAddGP substantially underestimated the amplitude of the periodic component and long-term trend.

Figure 2 shows these findings generalize as $N$ and $D$ increase. As the posterior of these high-dimensional models cannot be easily visualized, we quantified model performance based on the coverage of posterior 95% intervals with respect to the true function decomposition. Since both MultiAddGP and NAddGP are Bayesian models we do not expect that these intervals will cover

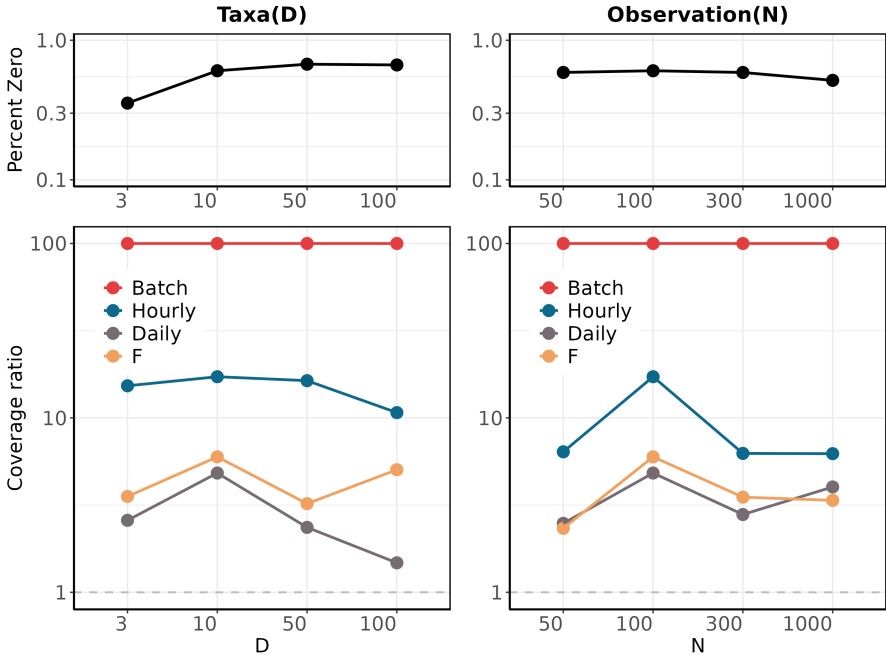

Figure 2: **At all tested dimensions $N$ and $D$, posterior intervals from MultiAddGPs cover the truth more frequently than NAddGPs.** The first row illustrates how data sparsity varied with dimensions $N$ and $D$ in our simulation studies. The second row shows the ratio between coverage of 95% Credible intervals from MultiAddGPs compared to NAddGPs. Each datapoint represents the mean over three simulations. The ratio is always positive illustrating MultiAddGPs cover the truth substantially more than NAddGPs. Coverage ratios for each of the decomposed components *Batch*, *Hourly*, and *Daily* and the cumulative function $F$ are shown. Note: This figure is also included in an extended version of this work currently under review for journal publication [5].

the truth with 95% probability. As a result, we focus on the ratio of coverage between the MultiAddGP and the NAddGP. Positive values of this coverage ratio indicate that the MultiAddGP model covers the truth more often than the NAddGP model. In all simulations, at all sample sizes $N$ and number of taxa $D$, the MultiAddGP models covered the truth more frequently than the NAddGP models.

## 4 Conclusion & Future Work

We have introduced MultiAddGPs, a Bayesian Multinomial Logistic-Normal additive regression model designed to address the statistical challenges of analyzing microbiota data. By incorporating recent advancements in Marginally Latent Matrix-t Processes (MLTPs), we developed computationally efficient inference methods, now implemented in the *fido* R package since version 1.1.0 [32]. Our simulations demonstrate that MultiAddGPs effectively disentangle linear and nonlinear effects, which highlight their potential for real-world applications. Looking ahead, our ongoing work focuses on applying MultiAddGPs to large-scale microbiome datasets to extract biologically meaningful insights and developing robust methods for hyperparameter selection, such as optimizing kernel parameters.

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

# A  Review of Marginally Latent Matrix-t Process and the CU Sampler

**Notation.** In this article, we denote matrix and vector dimensions with unbolded uppercase letters (e.g., $N$), matrices and matrix-valued functions using bold uppercase symbols (e.g., $\boldsymbol{X}$), vectors and vector-valued functions with bold lowercase symbols (e.g., $\mathbf{x}$), and scalars and scalar functions as unbolded lowercase symbols (e.g., $x$). For matrices, we index specific rows as $\mathbf{X}_{d\cdot}$ and columns as $\mathbf{X}_{\cdot n}$. We denote vector-valued stochastic processes using the same notation as matrices (e.g. $\boldsymbol{Y}$) since, in practice, we only evaluate these at a finite number of test points.

We describe the class of Marginally Latent Matrix-T Processes (MLTPs) by sequentially generalizing from Matrix-T Processes to Latent Matrix-T Processes (LTPs) and finally MLTPs. We then describe the subset of Bayesian Multinomial Logistic-Normal MLTPs (MLN-MLTPs) before reviewing the inference of this class of models.

## A.1  Defining Marginally Latent Matrix-T Processes (MLTPs)

Just as Gaussian processes can be defined based on the marginal properties of the multivariate normal, matrix normal processes and Matrix-T processes can be defined by the marginal properties of the matrix normal and matrix-T distributions [32]. Matrix-T processes generalize Student-T processes and Gaussian processes [32].

*Definition* A.1 (Matrix-T Process). A stochastic process $\mathbf{Y} \sim TP(\nu, \mathbf{M}, \mathbf{V}, \mathbf{A})$ defined on the set $\mathcal{W} = \mathcal{W}^{(1)} \times \mathcal{W}^{(2)}$ is a matrix-T process if $\mathbf{Y}$ evaluated on any two finite subsets $\mathcal{X}^{(1)} \subset \mathcal{W}^{(1)}$ and $\mathcal{X}^{(2)} \subset \mathcal{W}^{(2)}$ is a random matrix $\mathbf{Y}$ of dimension $|\mathcal{X}^{(1)}| \times |\mathcal{X}^{(2)}|$ that follows a matrix-T distribution: $\mathbf{Y} \sim T(\nu, \mathbf{M}, \mathbf{V}, \mathbf{A})$. $\nu$ is a scalar value strictly greater than zero. Let $x_i^{(1)}, x_j^{(1)} \in \mathcal{X}^{(1)}$ and $x_i^{(2)}, x_j^{(2)} \in \mathcal{X}^{(2)}$. $\mathbf{M}_{ij} = \mathbf{M}(x_i^{(1)}, x_j^{(2)})$ is the matrix function representing the mean, and $\mathbf{V}_{ij} = \mathbf{V}(x_i^{(1)}, x_j^{(1)})$ and $\mathbf{A}_{ij} = \mathbf{A}(x_i^{(2)}, x_j^{(2)})$ are kernel functions.

Latent Matrix-T Processes (LTP) generalize Matrix-T processes. $\mathbf{Y}$ is said to be an LTP if

$$\mathbf{Y} \sim g(\mathbf{\Pi}, \lambda)$$
$$\mathbf{\Pi} = \phi^{-1}(\mathbf{H})$$
$$\mathbf{H} \sim TP(\nu, \mathbf{M}, \mathbf{V}, \mathbf{A}).$$

where $g$ is any distribution depending on parameters $\mathbf{\Pi}$ as well as hyperparameters $\lambda$ and $\phi$ is a known transform. LTPs can alternatively be written as a joint model $p(\mathbf{Y}, \mathbf{H})$.

A stochastic process $\mathbf{Y}$ is Marginally LTP (MLTP) if it can be described by a joint distribution $p(\mathbf{Y}, \mathbf{H}, \mathbf{\Phi})$ with a marginal $p(\mathbf{Y}, \mathbf{H})$ that is a LTP. [32] showed that a wide variety of linear, dynamic linear, and non-linear regression models are MLTP. In Appendix B, we show that our proposed class of generalized additive Gaussian process regression models are MLTP as well.

## A.2  Bayesian Multinomial Logistic Normal MLTPs (MLN-MLTPs)

Bayesian Multinomial Logistic Normal MLTPs (MLN-MLTPs) are a subtype of MLTPs that are particularly useful for the analysis of microbiome data. In MLN-MLTPs, the distribution $g$ is a product multinomial: $p(\mathbf{Y}_{\cdot 1}, \dots, \mathbf{Y}_{\cdot N}) \sim \prod_{n=1}^{N} \text{Multinomial}(\mathbf{\Pi}_{\cdot n})$ and the transform $\phi$ is an invertible log-ratio transform from the $D$-dimensional simplex to $D-1$ dimensional real-space: $\mathbf{H}_{\cdot n} = \phi(\mathbf{\Pi}_{\cdot n} \in \mathbb{S}^D) \in \mathbb{R}^{D-1}$. Canonically, we used the following Additive Log-Ratio (ALR) transform which takes the $D$-th taxa as a reference:

$$\mathbf{H}_{\cdot n} = \phi(\mathbf{\Pi}_{\cdot n}) = \left\{ \log\left(\frac{\pi_{1n}}{\pi_{Dn}}\right), \dots, \log\left(\frac{\pi_{(D-1)n}}{\pi_{Dn}}\right) \right\}^T. \tag{5}$$

We choose this transform for computational efficiency as discussed in [32]. There is no loss in generality as posterior samples taken with respect to the $\text{ALR}_D$ coordinate system can be transformed into any other log-ratio coordinate system [25, Appendix A.3]. For context, the $\text{ALR}_D$ transform is the inverse of the softmax transform.

## A.3 Collapsed-Uncollapsed (CU) Sampler

The definition of MLTPs is key to efficient inference. If a model $p(\mathbf{Y}, \mathbf{H}, \mathbf{\Phi})$ has a closed-form marginal $p(\mathbf{Y}, \mathbf{H})$ that is an LTP, then its closed form conditional $p(\mathbf{\Phi} \mid \mathbf{Y}, \mathbf{H})$ likely exists. We call the marginal $p(\mathbf{Y}, \mathbf{H})$ the *collapsed form* and $p(\mathbf{\Phi} \mid \mathbf{Y}, \mathbf{H})$ the *uncollapsed form*. The posterior of an MLTP factors as

$$p(\mathbf{H}, \Phi \mid \mathbf{Y}) = p(\mathbf{\Phi} \mid \mathbf{H}, \mathbf{Y})p(\mathbf{H} \mid \mathbf{Y})$$

with the uncollapsed form as the first term and the posterior of the collapsed form as the second. As the collapsed form is rarely conjugate, techniques such as MCMC can be used to obtain samples from it's posterior. Then, conditioned on those samples, the uncollapsed form can be used to obtain samples from the joint posterior. Especially when $\mathbf{\Phi}$ is high-dimensional, this *Collapse-Uncollapse* sampler can be much more efficient than common alternatives [32]. Still, the most substantial enhancements occur when approximations to the collapsed form are considered.

We have developed a Laplace approximation for the collapsed form of MLTPs:

$$p(vec(\mathbf{H}) \mid \mathbf{Y}) \approx N(vec(\hat{\mathbf{H}}), \nabla^{-2}[vec(\hat{\mathbf{H}})])$$

where $\hat{\mathbf{H}}$ denotes the *Maximum A Posteriori (MAP)* estimate of the collapsed form and $\nabla^{-2}[vec(\hat{\mathbf{H}})]$ denotes the inverse Hessian of the collapsed form evaluated at the MAP estimate. To facilitate this approximation, we derived analytical results for the gradient and Hessian of the collapsed form [32]. Focusing on applications to MLN-MLTPs, we proved error bounds on the Laplace approximation and provided simulation and real analyses showing that the approximation was extremely accurate in the context of microbiome data analysis. Beyond the accuracy of posterior calculations, we showed that this CU sampler with the Laplace approximation (simply referred to as the CU sampler in the following text) was often 4-5 orders of magnitude faster than MCMC and 1-2 orders of magnitude faster than black-box variational inference while also being more accurate than the later. The CU sampler for MLN-MLTPs, along with uncollapse samplers for linear and non-linear regression models, is publicly available on CRAN as part of the *fido* software package.

# B Proof for MultiAddGPs are Marginal Latent Matrix-T Process (MLTPs)

## B.1 Derivation of Collapsed form

**Theorem B.1.** *If*

$$\mathbf{Y} \mid \mathbf{\Lambda} \sim MN(\mathbf{\Lambda}\mathbf{X}, \mathbf{\Sigma}, \mathbf{\Gamma})$$
$$\mathbf{\Lambda} \sim MN(\mathbf{\Theta}, \mathbf{\Sigma}, \mathbf{Z})$$

*and $\mathbf{\Sigma}$ is known, then the posterior of $\mathbf{\Lambda}$ is given by:*

$$\mathbf{\Lambda} \mid \mathbf{\Sigma}, \mathbf{Y} \sim MN\left((\mathbf{Y}\mathbf{\Gamma}^{-1}\mathbf{X}^T + \mathbf{\Theta}\mathbf{Z}^{-1})(\mathbf{X}\mathbf{\Gamma}^{-1}\mathbf{X}^T + \mathbf{Z}^{-1})^{-1}, \mathbf{\Sigma}, (\mathbf{X}\mathbf{\Gamma}^{-1}\mathbf{X}^T + \mathbf{Z}^{-1})^{-1}\right)$$

*Proof.* Using the density function of the matrix normal distribution, we can write:

$$\mathbf{\Lambda} \mid \mathbf{Y} \propto \exp\left[-\frac{1}{2}\text{tr}\left(\mathbf{\Sigma}^{-1}(\mathbf{Y} - \mathbf{\Lambda}\mathbf{X})\mathbf{\Gamma}^{-1}(\mathbf{Y} - \mathbf{\Lambda}\mathbf{X})^T\right)\right] \times \exp\left[-\frac{1}{2}\text{tr}\left(\mathbf{\Sigma}^{-1}(\mathbf{\Lambda} - \mathbf{\Theta})\mathbf{Z}^{-1}(\mathbf{\Lambda} - \mathbf{\Theta})^T\right)\right]$$

Combining the exponents and expanding the term:

$$\propto \exp\left[-\frac{1}{2}\text{tr}\left(\mathbf{\Sigma}^{-1}\left(\mathbf{Y}\mathbf{\Gamma}^{-1}\mathbf{Y}^T - \mathbf{\Lambda}\mathbf{X}\mathbf{\Gamma}^{-1}\mathbf{Y}^T - \mathbf{Y}\mathbf{\Gamma}^{-1}\mathbf{X}^T\mathbf{\Lambda}^T + \mathbf{\Lambda}\mathbf{X}\mathbf{\Gamma}^{-1}\mathbf{X}^T\mathbf{\Lambda}^T\right.\right.\right.$$
$$\left.\left.\left. + \mathbf{\Lambda}\mathbf{Z}^{-1}\mathbf{\Lambda}^T - \mathbf{\Theta}\mathbf{Z}^{-1}\mathbf{\Lambda}^T - \mathbf{\Lambda}\mathbf{Z}^{-1}\mathbf{\Theta}^T + \mathbf{\Theta}\mathbf{Z}^{-1}\mathbf{\Theta}^T\right)\right)\right].$$

$$\propto \exp\left[-\frac{1}{2}\text{tr}\left(\mathbf{\Sigma}^{-1}\left(-\mathbf{\Lambda}\mathbf{X}\mathbf{\Gamma}^{-1}\mathbf{Y}^T - \mathbf{Y}\mathbf{\Gamma}^{-1}\mathbf{X}^T\mathbf{\Lambda}^T + \mathbf{\Lambda}\mathbf{X}\mathbf{\Gamma}^{-1}\mathbf{X}^T\mathbf{\Lambda}^T + \mathbf{\Lambda}\mathbf{Z}^{-1}\mathbf{\Lambda}^T\right.\right.\right.$$
$$\left.\left.\left. - \mathbf{\Theta}\mathbf{Z}^{-1}\mathbf{\Lambda}^T - \mathbf{\Lambda}\mathbf{Z}^{-1}\mathbf{\Theta}^T\right)\right)\right]$$

Grouping like terms:

$$\boldsymbol{\Lambda} \mid \mathbf{Y} \propto \exp\left[-\frac{1}{2}\mathrm{tr}\left(\boldsymbol{\Sigma}^{-1}\left(\boldsymbol{\Lambda}(\mathbf{X}\boldsymbol{\Gamma}^{-1}\mathbf{X}^T + \mathbf{Z}^{-1})\boldsymbol{\Lambda}^T - \boldsymbol{\Lambda}(\mathbf{X}\boldsymbol{\Gamma}^{-1}\mathbf{Y}^T + \mathbf{Z}^{-1}\boldsymbol{\Theta}^T) - (\mathbf{Y}\boldsymbol{\Gamma}^{-1}\mathbf{X}^T + \boldsymbol{\Theta}\mathbf{Z}^{-1})\boldsymbol{\Lambda}^T)\right)\right]$$

$$\propto \exp\left(-\frac{1}{2}\mathrm{tr}\left\{\boldsymbol{\Sigma}^{-1}\left((\boldsymbol{\Lambda} - (\mathbf{Y}\boldsymbol{\Gamma}^{-1}\mathbf{X}^T - \boldsymbol{\Theta}\mathbf{Z}^{-1})(\mathbf{X}\boldsymbol{\Gamma}^{-1}\mathbf{X}^T + \mathbf{Z}^{-1})^{-1})\right.\right.$$
$$\left.\left.\times(\mathbf{X}\boldsymbol{\Gamma}^{-1}\mathbf{X}^T + \mathbf{Z}^{-1})\left(\boldsymbol{\Lambda} - (\mathbf{Y}\boldsymbol{\Gamma}^{-1}\mathbf{X}^T - \boldsymbol{\Theta}\mathbf{Z}^{-1})(\mathbf{X}\boldsymbol{\Gamma}^{-1}\mathbf{X}^T + \mathbf{Z}^{-1})^{-1}\right)^T\right)\right\}\right).$$

which implies that

$$\boldsymbol{\Lambda} \mid \boldsymbol{\Gamma}, \mathbf{Y} \sim \mathsf{MN}\left((\mathbf{Y}\boldsymbol{\Gamma}^{-1}\mathbf{X}^T - \boldsymbol{\Theta}\mathbf{Z}^{-1})(\mathbf{X}\boldsymbol{\Gamma}^{-1}\mathbf{X}^T + \mathbf{Z}^{-1})^{-1}, \boldsymbol{\Sigma}, (\mathbf{X}\boldsymbol{\Gamma}^{-1}\mathbf{X}^T + \mathbf{Z}^{-1})^{-1}\right)$$

$\square$

Note that in the special case where $\mathbf{X} = \mathbf{I}$, i.e., a model of the form:

$$\mathbf{Y} \mid \boldsymbol{\Lambda} \sim \mathsf{MN}(\boldsymbol{\Lambda}, \boldsymbol{\Sigma}, \boldsymbol{\Gamma})$$
$$\boldsymbol{\Lambda} \sim \mathsf{MN}(\boldsymbol{\Theta}, \boldsymbol{\Sigma}, \mathbf{Z})$$

then the above result simplifies to

$$\boldsymbol{\Lambda} \mid \boldsymbol{\Sigma}, \mathbf{Y} \sim \mathsf{MN}\left((\mathbf{Y}\boldsymbol{\Gamma}^{-1} + \boldsymbol{\Theta}\mathbf{Z}^{-1})(\boldsymbol{\Gamma}^{-1} + \mathbf{Z}^{-1})^{-1}, \boldsymbol{\Sigma}, (\boldsymbol{\Gamma}^{-1} + \mathbf{Z}^{-1})^{-1}\right).$$

### B.2 Derivation of Uncollapsed form

Here, we demonstrate how to efficiently compute and sample from the conditional posterior $p(\mathbf{F}, \mathbf{B}, \mathbf{f}^{(1)}, \ldots, \mathbf{f}^{(K)}, \boldsymbol{\Sigma}|\mathbf{H}, \mathbf{Y}, \mathbf{X}, \mathbf{Z})$. Since $\mathbf{F}$ and $\boldsymbol{\Sigma}$ are conditionally independent of $\mathbf{Y}$ given $\mathbf{H}$, and $\mathbf{B}, \mathbf{f}^{(1)}, \ldots, \mathbf{f}^{(K)}$ are conditionally independent of $\mathbf{H}$ given $\mathbf{F}$, by applying the chain rule, we can rewrite the equation as:

$$p(\mathbf{F}, \mathbf{B}, \mathbf{f}^{(1)}, \ldots, \mathbf{f}^{(K)}, \boldsymbol{\Sigma}|\mathbf{H}, \mathbf{Y}, \mathbf{X}, \mathbf{Z}) = p(\mathbf{B}, \mathbf{f}^{(1)}, \ldots, \mathbf{f}^{(K)}|\mathbf{F}, \boldsymbol{\Sigma}, \mathbf{X}, \mathbf{Z})p(\mathbf{F}|\boldsymbol{\Sigma}, \mathbf{H}, \mathbf{X}, \mathbf{Z})p(\boldsymbol{\Sigma}|\mathbf{H}, \mathbf{X}, \mathbf{Z})$$

The second and third parts on the right-hand side of the equation represent the posterior of a multivariate conjugate linear model, which can be sampled efficiently from Appendix C of [32].

To sample from the first part of the equation, we developed *backsampling* algorithm. The idea is motivated by the back-fitting algorithm in the Generalized Additive Model. Specifically, given the samples from $\mathbf{F}$ and $\boldsymbol{\Sigma}$, we draw sample iteratively from $p(\mathbf{B}|\mathbf{F})$, $p(\mathbf{f}^{(1)}|\mathbf{F}, \mathbf{B})$, ..., $p(\mathbf{f}^{(K)}|\mathbf{F}, \mathbf{B}, \mathbf{f}^{(1)}, \ldots, \mathbf{f}^{(K-1)})$. Starting with $\mathbf{B}$, define $\mathbf{B}^* = \mathbf{F} - \sum_{j=1}^{K}\boldsymbol{\Theta}^{(j)}(\mathbf{Z}^{(j)})$, then we can write:

$$\mathbf{B}^* \sim MN(\mathbf{B}\mathbf{X}, \boldsymbol{\Sigma}, \boldsymbol{\Gamma}^*)$$
$$\mathbf{B} \sim N(\boldsymbol{\Theta}^{(0)}, \boldsymbol{\Sigma}, \boldsymbol{\Gamma}^{(0)})$$

where $\boldsymbol{\Gamma}^* = \sum_{j=1}^{K}\boldsymbol{\Gamma}^{(j)}$, As the above model is a matrix conjugate linear model, we can sample from its closed form:

$$\mathbf{B}|\mathbf{B}^*, \boldsymbol{\Sigma} \sim MN((\mathbf{B}^*\boldsymbol{\Gamma}^{-*}X^T + \boldsymbol{\Theta}^{(0)}\boldsymbol{\Gamma}^{-(0)})(X\boldsymbol{\Gamma}^{-*}X^T + \boldsymbol{\Gamma}^{-(0)})^{-1}, \boldsymbol{\Sigma}, (X\boldsymbol{\Gamma}^{-*}X^T + \boldsymbol{\Gamma}^{-(0)})^{-1})$$

where $\boldsymbol{\Gamma}^{-*}$ and $\boldsymbol{\Gamma}^{-(0)}$ are short-hand for $(\boldsymbol{\Gamma}^*)^{-1}$ and $(\boldsymbol{\Gamma}^{(0)})^{-1}$ respectively.

We then use a similar process for $f^{(k)}$. Define $\mathbf{f}^* = \mathbf{F} - \mathbf{B}\mathbf{X} - \sum_{i=1}^{k-1}\mathbf{f}^{(i)} - \sum_{j=k+1}^{K}\boldsymbol{\Theta}^{(j)}(\mathbf{Z}^{(j)})$. Then we can use a similar process to sample for $f^{(k)}$:

$$\mathbf{f}^* \sim MN(\mathbf{f}^{(k)}, \boldsymbol{\Sigma}, \boldsymbol{\Gamma}^*)$$
$$\mathbf{f}^{(k)} \sim N(\boldsymbol{\Theta}^{(k)}, \boldsymbol{\Sigma}, \boldsymbol{\Gamma}^{(k)})$$

where $\mathbf{\Gamma}^* = \sum_{j=k+1}^{K} \mathbf{\Gamma}^{(j)}$ and we can sample from its closed-form conditional distribution:

$$\mathbf{f}^{(k)} \mid \mathbf{\Sigma}, \mathbf{f}^* \sim N\left(\left[\mathbf{f}^*\mathbf{\Sigma}^{-*} + \mathbf{\Theta}^{(k)}\mathbf{\Gamma}^{-(k)}\right]\left[\mathbf{\Gamma}^{-*} + \mathbf{\Gamma}^{-(k)}\right]^{-1}, \mathbf{\Sigma}, \left[\mathbf{\Gamma}^{-*} + \mathbf{\Gamma}^{-(k)}\right]^{-1}\right)$$

where $\mathbf{\Gamma}^{-*}$ and $\mathbf{\Gamma}^{-(k)}$ are short-hand for $(\mathbf{\Gamma}^*)^{-1}$ and $(\mathbf{\Gamma}^{(k)})^{-1}$ respectively. Finally, we set

$$\mathbf{f}^{(K)} = \mathbf{F} - \mathbf{BX} - \sum_{k=1}^{K-1} \mathbf{f}^{(k)}$$

## C   Pseudo code of Extended Collapse-Uncollapsed Sampler

In this section, we first present the pseudo-code for the Back Sampler (BS), which efficiently samples $\mathbf{B}$ and $\mathbf{f}^k$ for $k \in 1, \ldots, K$. Following this, we provide the full pseudo-code for the extended Collapse-Uncollapsed (CU) sampler designed for MultiAddGPs models. Note that in algorithm 2, the sampler from step 3-4 can be found in the [32] Appendix C.

---

**Algorithm 1** Back Sampler (BS)

---

1: **Input:** $\{\mathbf{Y}, \mathbf{X}, \mathbf{Z}\}$ are data observation, $\{\mathbf{F}, \mathbf{\Sigma}\}$ are samples from CU sampler , $\Lambda = \{\mathbf{\Theta}^{(0)}, \ldots, \mathbf{\Theta}^{(k)}, \mathbf{\Gamma}^{(0)}, \ldots, \mathbf{\Gamma}^{(k)}\}$ is a set of prior input
2: **Output:** $S$ samples of the form $(\mathbf{B}, \mathbf{f}^{(k)}, k \in \{1, \ldots, K\})$
3: **for** $s = 1$ to $S$ **do**
4:     $\mathbf{B}^* = \mathbf{F} - \sum_{j=1}^{K} \mathbf{\Theta}^{(j)}(\mathbf{Z}^{(j)})$
5:     $\mathbf{\Gamma}^* = \sum_{j=1}^{K} \mathbf{\Gamma}^{(j)}$
6:     Sample $\mathbf{B}|\mathbf{B}^*, \mathbf{\Sigma} \sim MN((\mathbf{B}^*\mathbf{\Gamma}^{-*}\mathbf{X}^T + \mathbf{\Theta}^{(0)}\mathbf{\Gamma}^{-(0)})(\mathbf{X}\mathbf{\Gamma}^{-*}\mathbf{X}^T + \mathbf{\Gamma}^{-(0)})^{-1}, \mathbf{\Sigma}, (\mathbf{X}\mathbf{\Gamma}^{-*}\mathbf{X}^T + \mathbf{\Gamma}^{-(0)})^{-1})$ where $\mathbf{\Gamma}^{-*}$ and $\mathbf{\Gamma}^{-(0)}$ are short-hand for $(\mathbf{\Gamma}^*)^{-1}$ and $(\mathbf{\Gamma}^{(0)})^{-1}$ respectively.
7:     **for** $j = 1$ to $K$ **do**
8:         **if** j = 1, ..., $k - 1$ **then**
9:             $\mathbf{f}^* = \mathbf{F} - \mathbf{BX} - \sum_{i=1}^{k-1} \mathbf{f}^{(i)} - \sum_{j=k+1}^{K} \mathbf{\Theta}^{(j)}(\mathbf{Z}^{(j)})$
10:            $\mathbf{\Gamma}^* = \sum_{j=k+1}^{K} \mathbf{\Gamma}^{(j)}$
11:            Sample $\mathbf{f}^{(k)} \mid \mathbf{\Sigma}, \mathbf{f}^* \sim N\left(\left[\mathbf{f}^*\mathbf{\Sigma}^{-*} + \mathbf{\Theta}^{(k)}\mathbf{\Gamma}^{-(k)}\right]\left[\mathbf{\Gamma}^{-*} + \mathbf{\Gamma}^{-(k)}\right]^{-1}, \mathbf{\Sigma}, \left[\mathbf{\Gamma}^{-*} + \mathbf{\Gamma}^{-(k)}\right]^{-1}\right)$
                where $\mathbf{\Gamma}^{-*}$ and $\mathbf{\Gamma}^{-(k)}$ are short-hand for $(\mathbf{\Gamma}^*)^{-1}$ and $(\mathbf{\Gamma}^{(k)})^{-1}$ respectively.
12:        **else**
13:            Sample $\mathbf{f}^{(K)} = \mathbf{F} - \mathbf{BX} - \sum_{k=1}^{K-1} \mathbf{f}^{(k)}$
14:        **end if**
15:    **end for**
16: **end for**
17: **return** $\mathbf{B}, \mathbf{f}^{(k)}$

---

---

**Algorithm 2** The Collapse-Uncollapse (CU) Sampler for **MultiAddGPs** Models

---

1: **Input:** $\{\mathbf{Y}, \mathbf{X}, \mathbf{Z}\}$ are data observation, $\Delta = \{\Lambda, \mathbf{\Xi}, \nu\}$ is a set of prior input
2: **Output:** $S$ sample of $\{\mathbf{H}, \mathbf{\Sigma}, \mathbf{F}, \mathbf{B}, \mathbf{f}^{(k)}, k \in \{1, \ldots, K\}\}$
3: Sample $S$ of $\mathbf{H} \sim p(\mathbf{H}|\mathbf{Y}, \mathbf{X}, \mathbf{Z}, \Delta)$ where $p(\mathbf{H}|\mathbf{Y}, \mathbf{X}, \mathbf{Z}, \Delta)$ is an LTP;
4: Sample $S$ of $\mathbf{\Sigma} \sim p(\mathbf{\Sigma}|\mathbf{H}, \mathbf{X}, \mathbf{Z})$;
5: Sample $S$ of $\mathbf{F} \sim p(\mathbf{F}|\mathbf{H}, \mathbf{\Sigma}, X, Z)$;
6: Sample $S$ of $\mathbf{B}, \mathbf{f}^{(k)} = \text{BS}(\mathbf{Y}, \mathbf{X}, \mathbf{Z}, \mathbf{F}, \mathbf{\Sigma}, \Lambda)$

---

## D   Simulation Study

To evaluate the implementation and investigate the behavior of the MultiAddGPs model, we simulated a synthetic microbial community time-series comprising four bacterial taxa across 600

time points, based on the following model:

$$\mathbf{Y}_{.n} \sim \text{Multinomial}(\mathbf{\Pi}_{.n})$$
$$\mathbf{\Pi}_{.n} = ALR^{-1}(\mathbf{H})$$
$$\mathbf{F}_{.n} = 2.7 + 3x_n^{(batch)} + \mathbf{f}^{(periodic)}(t_n) + \mathbf{f}^{(trend)}(t_n)$$
$$\mathbf{f}^{(trend)}(t_n) \sim MN(0, \mathbf{\Sigma}, \mathbf{\Gamma}^{(trend)})$$
$$\mathbf{f}^{(periodic)}(t_n) \sim MN(0, \mathbf{\Sigma}, \mathbf{\Gamma}^{(periodic)})$$

Here, we set $\mathbf{\Sigma}$ as a covariance matrix with off-diagonal elements of 0.9 and diagonal elements of 1.5. The periodic kernel is defined as $\mathbf{\Gamma}^{(periodic)} = 4 \exp\left(-\frac{2\sin^2\left(\frac{\pi|t-t'|}{25}\right)}{30^2}\right)$, while the trend kernel is modeled as $\mathbf{\Gamma}^{(trend)} = \exp\left(-\frac{(t-t')^2}{2\times30^2}\right)$. After obtaining the posterior samples from the MultiAddGPs model, we apply a sum-to-zero constraint to facilitate model identification.

In Figure 1 of the main text, we illustrate the model's ability to successfully decompose the simulated microbiome time-series for a single taxon. In Figures 3 and 4 below, we further demonstrate this decomposition for two additional taxa.

Next, we assessed the scalability of the model. However, as the dimensions ($D$) and number of time points ($N$) increased, it became increasingly challenging to simulate data with a distinct non-linear trend suitable for additive modeling. To address this, we replaced the non-linear trend kernel $\mathbf{\Gamma}^{(trend)}$ with a linear kernel: $\mathbf{\Gamma}^{(trend)} = 20^2 + (t-c)(t'-c)$, while keeping the rest of the model unchanged. We then simulated this modified model across various combinations of $D$ and $N$, where $D \in 3, \ldots, 100$ and $N \in 20, \ldots, 1000$. For each combination of $(D, N)$, we generated three simulated datasets. The coverage ratio, presented in Figure 2 below, represents the average across these three simulations.

Analysis of the simulated dataset revealed that the estimates for the unobserved compositions, $\mathbf{H}$, and latent factors, $\mathbf{F}$, obtained from the MultiAddGPs model were more accurate compared to those derived from the standard approach of normalizing read counts to proportions (NAddGPs). Furthermore, our model successfully disentangled distinct effects arising from multiple linear and non-linear factors. These results suggest that our model is capable of effectively decomposing longitudinal microbiota data into a mixture of linear and non-linear additive components.

All implements were compiled and run using gcc version 9.1.0 and R version 4.3.2. All replicates of the simulated count data were supplied to the various implementations independently and the models were fit on identical hardware, allotted 64GB RAM, 4 cores, and restricted to a 48-hour upper limit on run-time. All code required to reproduce the results of the this article is available at https://github.com/Silverman-Lab/MultiAddGPs.

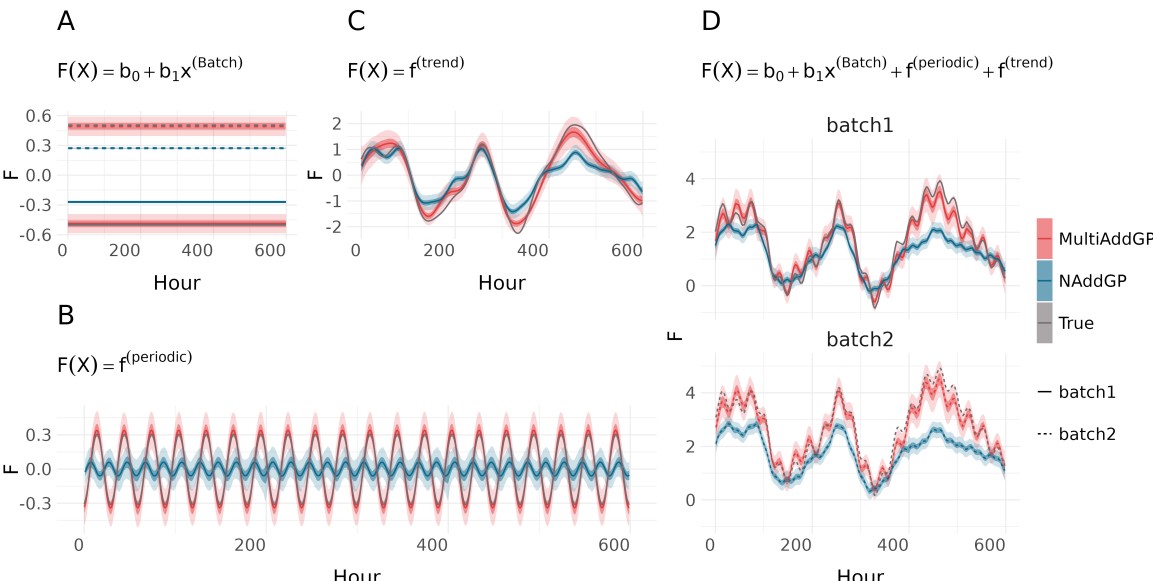

Figure 3: MultiAddGPs successfully decompose simulated microbiome time-series on Taxa 2. Note: This figure is also included in an extended version of this work currently under review for journal publication [5].

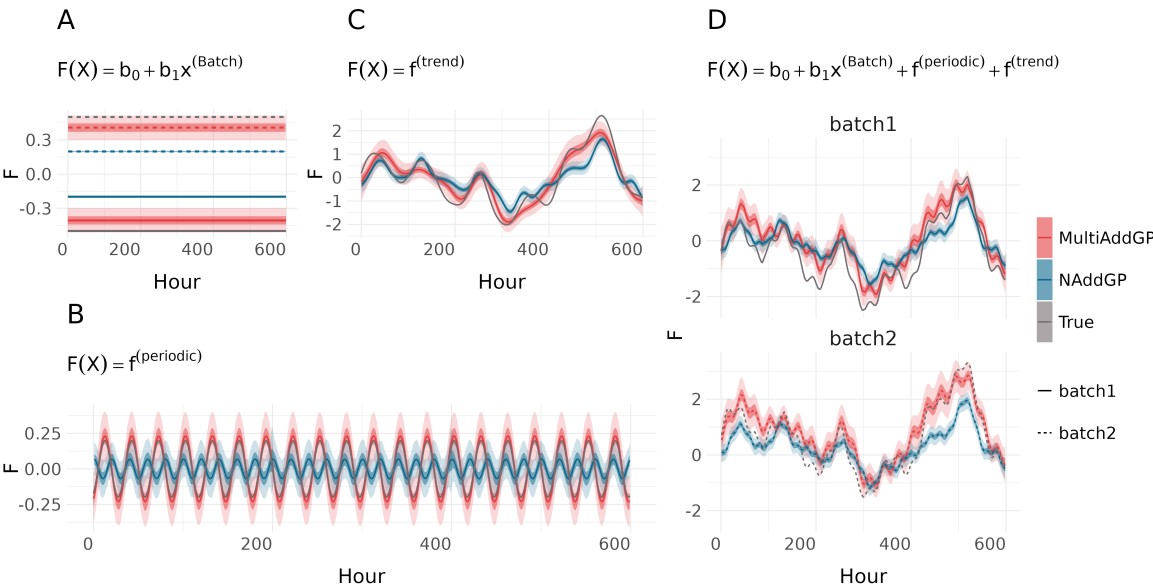

Figure 4: MultiAddGPs successfully decompose simulated microbiome time-series on Taxa 3. Note: This figure is also included in an extended version of this work currently under review for journal publication [5].

