# OpenReview forum: "Efficient Bayesian Additive Regression Models For Microbiome Studies"
_NeurIPS.cc/2024/Workshop/BDU — NeurIPS BDU Workshop 2024 Poster_

### Official Review · Reviewer_L5iz · 2024-09-26
**This is a nice paper that improves the MLTP approach, which implies a complex covariance structure, by adding the deconvoluted overlapping linear and non-linear effects. The earlier approach of MLTP either can have a non-linear effect or multiple disjoint linear effects. The motivation of the paper is to improve the estimation of the linear and non-linear effects of covariates on microbial or gene compositions.**

**Rating:** 7
**Confidence:** 4

**Review:**

1. My biggest problem with the paper is how the authors clearly reveal their identity by mentioning “we” and referring to their previous work in the reference.
2. The paper could be of more value if it were compared to mixed models since those also include the additive concept. And it could be that Mixed Models are simpler and better suited.
3. I am a bit unsure if the simulated data is generated in a fair manner, but at least it shows that some logic is working in their method.
4. The paper sounds really powerful until I read the algorithm, seeing that we actually have observable variables signaling clearly what is linear and non-linear. This makes the whole paper less useful as, in many biological applications, we do not have access to this level of information, and we merely have one observation, which is the result of collided linear and nonlinear covariates. I may be misunderstanding this last point here but if it is sensible I am hoping to be useful for the reader of this comment.
5. Finally, I think this work is absolutely valuable and sound and it should be accepted.

---

### Official Review · Reviewer_54z2 · 2024-09-26
**The paper titled "Efficient Bayesian Additive Regression Models For Microbiome and Gene Expression Studies" presents a framework for analyzing complex sequence count data, particularly within microbiome and gene expression contexts.**

**Rating:** 6
**Confidence:** 4

**Review:**

The authors address key challenges inherent in complex sequence count data, particularly its compositional and count-based nature, which traditional models often overlook by treating the data as Gaussian. By introducing the MultiAddGP models within a Bayesian Multinomial Logistic-Normal (MLN) framework, the paper extends upon prior methods like generalized linear models and Gaussian process models, allowing for both linear and non-linear effects to be accounted for simultaneously.

Strengths:
-The CU sampler speeds up inference, making it much faster than traditional methods like HMC, improving scalability for large datasets.
-The MultiAddGP model can capture both linear and non-linear effects, allowing for more accurate modeling of complex microbiome and gene expression data, which often have overlapping factors.

Limitations:
-The model's reliance on the Laplace approximation may not perform well in cases with low-sequencing depth, requiring alternative methods in such situations.
-While powerful for microbiome and gene expression data, the model’s focus on compositional count data may make it less applicable for simpler datasets.

NOTE: There is a link in the paper which is breaking the anonymity rule.

---

### Decision · Program_Chairs · 2024-10-09

Accept (Poster)